# 🧩RF Prior: Preserving Global-Context Priors for Efficient Instance Segmentation Transfer

## Abstract

We present an efficient transfer-learning framework that reparameterizes a state-of-the-art detector backbone—instantiated with a YOLO-family model—for polygon-based instance segmentation. Our key idea is a Receptive-Field Prior: the largest-receptive-field block (P5) of the backbone, pretrained for detection, is kept *fixed* to preserve global object context, while intermediate low-level blocks (P3–P4) are fine-tuned for boundary precision. We formalize this with a block-diagonal Gaussian prior on backbone weights, yielding a *MAP* objective that acts as implicit adaptation. Multi-scale features from P3–P5 are fused in a attentive decoder to predict per-instance polygons. Experiments show strong and stable performance compared with scratch training or naïve tuning strategy. This approach[1] highlights that carefully constrained reuse of high-level detector features—guided by an explicit inductive bias—can yield strong segmentation.

## 1 Introduction

Inductive bias—architectural constraints that shape the hypothesis space—is a principal driver of generalization in vision models. Classic CNNs hard–code translation equivariance and locality, while recent hybrids interleave convolution and attention to couple fine detail with global scene context (Liu et al., 2021; Wang et al., 2018; Liu et al., 2022; Woo et al., 2023). Such designs yield pyramidal hierarchies whose high–resolution stages capture boundaries and textures, and whose low–resolution stages aggregate semantics over large receptive fields. These hierarchies transfer well across tasks: detector backbones provide geometry–aware mid–level cues and dense heads refine them into pixel–accurate segmentations (Kirillov et al., 2019; Cheng et al., 2022). Yet *how* to preserve useful priors during fine–tuning remains open. Full freezing curbs adaptation under domain shift; full fine–tuning expands the search space, slows convergence, and can overwrite global context (Xuhong et al., 2018).

**Premise.** We observe that the largest receptive–field block (P5) of modern detectors—already enhanced with efficient attention in YOLOv12 (Tian et al., 2025)—encodes stable scene–level structure that is especially valuable for polygonal instance segmentation. We therefore *anchor* global semantics by freezing P5 and adapt only P3–P4, multi–scale decoder that performs a *single* area–restricted fusion at the P5→P4 interface. This early–adaptive, context–aware recipe sharpens boundaries, reduces texture overfitting, and accelerates convergence.

**Contributions.**

- **Receptive–Field Prior.** We cast transfer as *MAP* with a block–diagonal Gaussian over backbone weights (Sec. 3.2), unifying a $\delta$–prior on P5 (hard freezing) with zero–mean decay on adaptable blocks in a single objective Eq. 2.
- **Targeted Global-to-Local Fusion.** We introduce a multi–scale decoder that concentrates area–restricted attention once at the P5-to-P4 fusion, while keeping the context-aligned fusion; this focuses long–range cues exactly where mid–level features benefit most.
- **Automatic BBox-to-Polygon Mining for Transfer.** To leverage box–only corpora under background/label shift, we propose a simple mining module that converts detector boxes

---

[1]Our framework (code & dataset) will be released upon acceptance as Ultralytics-compatible pipeline.

into polygon pseudo–masks via candidate segmentation, multi–metric ranking, and contour simplification (Sec. 3.5); integrating these pseudo–polygons into our RF–prior pipeline yields further gains in boundary metrics with little to no inference overhead.

## 2 RELATED WORK

### 2.1 INDUCTIVE BIAS AND TRANSFER REGULARIZATION

Pyramidal backbones encode complementary scales by design; preserving their semantics during fine–tuning is key for generalization. $L^2$–SP contracts parameters toward source weights and mitigates catastrophic drift (Xuhong et al., 2018; Chen & Liu, 2022), while subnetwork freezing is used to retain global attention patterns in large–vision models (e.g., ViT–R (Zhai et al., 2022)). Our work adopts a *MAP* view in which a block–diagonal prior fixes the top semantic block (P5) via a $\delta$–prior and regularizes adaptable blocks with zero–mean decay, balancing stability and capacity (Sec. 3.2) More information of cross-task representation reuse are included in Appendix§A.1.

### 2.2 GLOBAL CONTEXT FOR DENSE PREDICTION

Non–local operators (Wang et al., 2018), criss–cross attention (Huang et al., 2019), and transformer (SETR (Zheng et al., 2021), SegFormer (Xie et al., 2021), DETR-like methods (Li et al., 2023)) inject global context but can be costly at high resolution. Area–restricted attention from the YOLOv12 family (Tian et al., 2025) offers a compute–aware alternative. Placing a single attention site at the stride–32 to 16 fusion is a targeted compromise that preserves long–range cues while keeping the highest–resolution stage lightweight (Sec. 3.3). We introduce, in Sec. 3.2–3.4, a prediction framework that leverages a prior structure and decoder to enable context-aligned interactions and implicit (model $\leftrightarrow$ latent space) optimization using decoder-coupled weight decay $\xi$.

### 2.3 AUTOMATIC BOX-TO-POLYGON PRIORS

Weakly and box–supervised segmentation has long converted coarse boxes into usable mask supervision via proposal mining and regularization, with BoxSup (Dai et al., 2015) and the "Simple Does It" line (Khoreva et al., 2017) as early milestones, and instance–level formulations such as BoxInst (Tian et al., 2021). Promptable segmenters like SAM (Kirillov et al., 2023) enable mask proposals from box prompts, while vision–language models such as CLIP (Radford et al., 2021) provide semantic filtering to favor class–consistent candidates. We situate our approach within this literature by using automatically mined polygons—obtained through proposal selection and contour simplification—as additional priors during transfer. Unlike heavy multi–stage pipelines, our integration couples mined polygons with an RF PRIOR and a Attentive Decoder, emphasizing boundary quality while retaining throughput.

## 3 METHODOLOGY

### 3.1 PRELIMINARIES

*Notation.* Given $x \in \mathbb{R}^{3 \times H \times W}$, the backbone $B_{\boldsymbol{\theta}_{\text{bb}}}$ produces a feature pyramid $\{F_3, F_4, F_5\}$ at strides $\{8, 16, 32\}$. We decompose $\boldsymbol{\theta}_{\text{bb}} = [\boldsymbol{\theta}_b; \boldsymbol{\theta}_5]$ and *freeze* $\boldsymbol{\theta}_5$ to preserve large–receptive–field semantics inherited from detection; $\boldsymbol{\theta}_b$ (P3/P4) and the decoder $H_{\boldsymbol{\theta}_{\text{seg}}}$ are optimized for segmentation. The head upsamples and fuses $\{F_3, F_4, F_5\}$ to logits $\hat{M} \in \mathbb{R}^{C \times H \times W}$, with masks $M = \sigma(\hat{M})$. *Intuition.* $F_5$ supplies global layout/category priors, while lower stages sharpen boundaries and local geometry.

### 3.2 RECEPTIVE-FIELD PRIOR

Let $\boldsymbol{\theta}_{\text{bb}} = [\boldsymbol{\theta}_3; \boldsymbol{\theta}_4; \boldsymbol{\theta}_5]$ denote backbone blocks (P3–P5) producing $F_\ell = B_{\boldsymbol{\theta}_\ell}(F_{\ell-1})$ with receptive-field radii $R_3 < R_4 < R_5$. We freeze P5 to the detector initialization $\boldsymbol{\theta}_{0,5}$ and fine-tune P3/P4 with SGD (with momentum; L2 weight decay). From a *MAP* perspective, this induces a *block–diagonal* prior (Fig. 1-a):

$$p(\boldsymbol{\theta}_{\text{bb}}) \propto \exp\left( -\frac{1}{2} \begin{bmatrix} \boldsymbol{\theta}_3 \\ \boldsymbol{\theta}_4 \end{bmatrix}^\top \text{diag}(\xi_3 I, \xi_4 I) \begin{bmatrix} \boldsymbol{\theta}_3 \\ \boldsymbol{\theta}_4 \end{bmatrix} \right) \delta(\boldsymbol{\theta}_5 - \boldsymbol{\theta}_{0,5}), \tag{1}$$

where $\xi_{3,4}$ coincide with the L2 weight–decay coefficients on P3/P4, and the delta factor encodes the hard freeze of $P5$ at $\boldsymbol{\theta}_{0,5}$.

Figure 1: Overview of the proposed transfer-learning scheme. A YOLOv12-x backbone produces multi-scale features P3–P5, where P5 (blue) is attention-augmented with the largest receptive field and is frozen as the RF PRIOR. During segmentation fine-tuning, only P3/P4 (gray) and the segmentation head receive $\nabla \mathcal{L}_{\text{seg}}$ updates. The segmentation head fuses P3, P4, and the fixed P5 context to predict instance masks.

**Objective.** On $\mathcal{D}_{\text{seg}} = \{(x_i, y_i)\}_{i=1}^{N}$ we minimize

$$\mathcal{L}_{\text{CC}}(\boldsymbol{\theta}) \;=\; \sum_{i=1}^{N} \underbrace{\ell_{\text{seg}}\big(H_{\boldsymbol{\theta}_{\text{seg}}} \circ B_{\boldsymbol{\theta}_{\text{bb}}}(x_i),\, y_i\big)}_{\text{Unified loss in Sec. 3.6}} \;+\; \frac{\xi_3}{2}\,\|\boldsymbol{\theta}_3\|_2^2 \;+\; \frac{\xi_4}{2}\,\|\boldsymbol{\theta}_4\|_2^2, \qquad \text{s.t. } \boldsymbol{\theta}_5 = \boldsymbol{\theta}_{0,5}. \quad (2)$$

**Update dynamics.** For block $\ell \in \{3, 4\}$ with step size $\eta_\ell$, we employ SGD with momentum coefficient $\mu \in [0, 1)$ and maintain a velocity $v_\ell$:

$$v_\ell^{t+1} = \mu\, v_\ell^t \;+\; g_\ell^t \;+\; \xi_\ell\, \boldsymbol{\theta}_\ell^t, \tag{3}$$

$$\boldsymbol{\theta}_\ell^{t+1} = \boldsymbol{\theta}_\ell^t \;-\; \eta_\ell\, v_\ell^{t+1}, \qquad g_\ell^t = \nabla_{\boldsymbol{\theta}_\ell} \ell_{\text{seg}}(\boldsymbol{\theta}^t), \tag{4}$$

while the hard constraint yields $\boldsymbol{\theta}_5^{t+1} = \boldsymbol{\theta}_{0,5}$. Writing $\boldsymbol{\theta}_\ell = \boldsymbol{\theta}_{0,\ell} + \Delta_\ell$ (detector init + change) makes the forgetting bias explicit:

$$\Delta_\ell^{t+1} \;=\; (1 - \eta_\ell \xi_\ell)\, \Delta_\ell^t \;-\; \eta_\ell \Big(\mu\, v_\ell^t + g_\ell^t + \xi_\ell\, \boldsymbol{\theta}_{0,\ell}\Big). \tag{5}$$

Thus, standard L2 weight decay *does not* preserve $\boldsymbol{\theta}_{0,\ell}$; it shrinks both the initialization and the task-driven change toward 0, whereas the P5 freeze preserves global semantics exactly ($\Delta_5 = 0$). *(If Nesterov momentum is used, replace $g_\ell^t$ in the first line with $\nabla_{\boldsymbol{\theta}_\ell} \ell_{\text{seg}}(\boldsymbol{\theta}^t - \eta_\ell \mu v_\ell^t)$; the rest remains analogous.)*

**Receptive-field locality and polygon adaptation.** Backpropagated gradients into block $\ell$ aggregate supervision from a spatial neighborhood $\mathcal{N}_{R_\ell}(x)$:

$$g_\ell^t \;\approx\; \sum_{u \in \mathcal{N}_{R_\ell}(x)} J_\ell^t(x, u)^\top \frac{\partial \ell_{\text{seg}}}{\partial F_\ell^t(u)}, \tag{6}$$

with $J_\ell$ the feature Jacobian. Since $R_3 < R_4 \ll R_5$ and $\boldsymbol{\theta}_5$ is frozen, updates concentrate on P3/P4 near polygon boundaries where $\partial \ell / \partial F_\ell$ is large. Eq. 5 then yields *compact yet plastic* adjustments in P3/P4 (via the factor $(1 - \eta_\ell \xi_\ell)$).

*Context modulation under a frozen P5.* Although $\boldsymbol{\theta}_5$ is frozen, the P5 block applies *intra-scale* area attention in an input-conditioned manner. Let

$$\widehat{F}_5(x) = B_{\boldsymbol{\theta}_5}^{\text{pre}}\big(F_4(x)\big), \qquad A(x) = \mathcal{A}_{\text{self}}\big(\widehat{F}_5(x); \boldsymbol{\theta}_{\mathcal{A}}\big) \in [0, 1]^{H/32 \times W/32},$$

and define the P5 *backbone output* as

$$F_5(x) = A(x) \odot \widehat{F}_5(x),$$

with $\boldsymbol{\theta}_5$ and $\boldsymbol{\theta}_{\mathcal{A}}$ held fixed during training (no parameter updates). Because both $A$ and $\widehat{F}_5$ depend on the input $x$ (and on $F_4$, which adapts via $\boldsymbol{\theta}_{3,4}$),

$$\frac{\partial F_5(x)}{\partial x} \;=\; \frac{\partial A}{\partial \widehat{F}_5}\frac{\partial \widehat{F}_5}{\partial x} \odot \widehat{F}_5 \;+\; A \odot \frac{\partial \widehat{F}_5}{\partial x} \;\neq\; 0.$$

Moreover, across training steps $t$, the evolving lower blocks imply

$$F_5^{t+1}(x) \;-\; F_5^t(x) \;\neq\; 0 \qquad \text{even though } \boldsymbol{\theta}_5, \boldsymbol{\theta}_{\mathcal{A}} \text{ remain fixed.}$$

The decoder then consumes $(F_5, F_4, F_3)$; cross-scale fusion therein aligns the P5 context with the evolving lower stages and the head.

### 3.3 MULTI-SCALE ATTENTIVE DECODER

We propose a decoder that adapts the global context encoded by our RF PRIOR to local evidence. This design strengthens cross-scale interaction, enabling more effective fusion of prior-driven global cues with local features (see Figure 1-b,c).

**Near-global context via SPPF.** We adopt the expansion a stride–preserving SPPF (Jocher, 2023) on $F_5$ yields $S = [\mathcal{P}_k^{(0)}(F_5)\|\mathcal{P}_k^{(1)}(F_5)\|\mathcal{P}_k^{(2)}(F_5)\|\mathcal{P}_k^{(3)}(F_5)]$, with $\mathcal{P}_k^{(0)} \equiv \mathrm{Id}$ and $\mathcal{P}_k^{(\ell)} = \mathcal{P}_k \circ \mathcal{P}_k^{(\ell-1)}$; $\tilde{F}_5 = \phi_{1\times1}(S)$ reprojects to the decoder width. Stacking stride–1 pooling enlarges the effective RF additively, $R_L = 1 + L(k-1)$ (e.g., $k{=}5$, $L{=}3 \Rightarrow R_L{=}13$ at stride 32).

**C2f/A2C2f parametric form.** For $X \in \mathbb{R}^{B \times C_{\mathrm{in}} \times H \times W}$,

$$\mathrm{C2f}(X;\, c, r, s, g, e) \;=\; P_{1\times1}^{(c)}\Big(\mathrm{Cat}\big[X,\, \psi^{(1)}(X), \ldots, \psi^{(r)}(X)\big]\Big),$$

where each $\psi^{(i)}$ is a bottleneck with expansion $e$, groups $g$, and internal shortcut flag $s \in \{0,1\}$; $c$ is the output width and $r$ the repeat count. Let $U = \mathrm{C2f}(X; c, r, s, g, e)$. Area-restricted attention is then defined by

$$Q = UW_Q, \quad K = UW_K, \quad V = UW_V, \qquad W_Q, W_K, W_V \in \mathbb{R}^{C \times d},$$

and a partition $\{\mathcal{A}_r\}_{r=1}^a$ of the spatial grid, with

$$Y \;=\; \Big(\bigoplus_{r=1}^{a} \mathrm{softmax}\big(\tfrac{Q_{\mathcal{A}_r}K_{\mathcal{A}_r}^\top}{\sqrt{d}}\big)\, V_{\mathcal{A}_r}\Big)W_O, \qquad W_O \in \mathbb{R}^{d \times C}.$$

We write $\mathsf{A}_a(U) = Y$ and define the gated variant

$$\mathrm{A2C2f}(X;\, c, r, s, g, e, a, \gamma) \;=\; U \;+\; \gamma\, \mathsf{A}_a(U), \qquad \gamma \in \mathbb{R}^C.$$

**Top–down fusion with a single attention site.** Attention is enabled at the $F_5 \oplus F_4$ fusion; the subsequent high–resolution stage uses C2f (Jocher, 2023). With channel widths and repeats fixed to $(c_4, r_4, s_4) = (512, 2, 0)$ at stride 16 and $(c_3, r_3, s_3) = (256, 2, 0)$ at stride 8, we write

$$C_4 = \mathrm{Cat}\big[\uparrow_2(\tilde{F}_5),\, F_4\big], \quad \hat{C}_4 = P_{1\times1}^{(c_4)}(C_4), \quad G_4 = \mathrm{A2C2f}(\hat{C}_4;\, c_4, r_4, s_4,\, g{=}1,\, e{=}1,\, a{=}4,\, \gamma),$$

$$C_3 = \mathrm{Cat}\big[\uparrow_2(G_4),\, F_3\big], \quad \hat{C}_3 = P_{1\times1}^{(c_3)}(C_3), \quad G_3 = \mathrm{C2f}(\hat{C}_3;\, c_3, r_3, s_3,\, g{=}1,\, e{=}1). \tag{7}$$

*Complexity.* For $Y = \mathsf{A}_a(Z)$ with $N{=}HW$, area partitioning gives $\mathcal{O}(N^2/a)$ time/memory; we use $a{=}4$ at stride 16 to concentrate attention where global$\to$local alignment is most beneficial.

### 3.4 GLOBAL-TO-LOCAL GRADIENT FLOW

Let $W_q^{5\to4}$ denote the query projection drawing from $F_5$ when attending into $F_4$. Although $\boldsymbol{\theta}_5$ is frozen, the attention path is trainable and yields

$$\frac{\partial \ell_{\mathrm{seg}}}{\partial F_5} \;=\; \big(W_q^\top (A \odot \tfrac{\partial \ell_{\mathrm{seg}}}{\partial Z})\big) R,$$

with attention map $A$ and reshape $R$. *Interpretation.* The decoder aligns shallow features to the global template in $F_5$ by implicitly reducing $\mathcal{E} = \sum_k \big\|F_{5,k} - \phi(F_{4,k})\big\|^2 + \big\|F_{5,k} - \phi(F_{3,k})\big\|^2$, where $\phi$ is learned projection into the query-aligned space (see Figure 1-c).

## 3.5 AUTOMATIC BBOX-TO-POLYGON GENERATION

We verify the applicability of our transfer-learning framework to *real-shifted* (background, label) proposed data by polygonizing bbox-only annotations and using them in the transfer stage (Fig. 1-d,e). From a YOLO box $l = (c, \hat{c}_x, \hat{c}_y, \hat{w}, \hat{h})$, we denormalize to $B = [x_1, y_1, x_2, y_2]$ and prompt SAM to obtain candidates $\{M_k\}_{k=1}^{K}$. For each $M_k$, we compute $\text{IoU}_k = \frac{|M_k \cap \mathcal{B}|}{|M_k \cup \mathcal{B}|}$, $\text{Cover}_k = \frac{|M_k \cap \mathcal{B}|}{|M_k|}$, and an optional CLIP score $s_k^{\text{clip}} = \cos\big(f_{\text{img}}(I \odot M_k),\ f_{\text{text}}(t_c)\big)$. After per-metric min–max scaling, we rank with $S_k = \alpha \widetilde{\text{IoU}}_k + \beta \widetilde{s}_k^{\text{clip}} + \gamma \widetilde{\text{Cover}}_k$; if any $\text{IoU}_k \geq \tau$ we take the pixelwise union $\bigvee_{k:\text{IoU}_k \geq \tau} M_k$, else we choose $\arg\max_k S_k$. The selected mask is polygonized via Douglas–Peucker with tolerance $\varepsilon$ (Ramer, 1972; Douglas & Peucker, 1973), vertices are clipped to $B$, re-normalized by $(W, H)$, and emitted as $[c, \hat{x}_1, \hat{y}_1, \ldots, \hat{x}_n, \hat{y}_n]$.

**Design rationale.** (1) *Multi-mask union.* SAM can partition a single object; an IoU gate ($\text{IoU} \geq \tau$) with pixelwise union consolidates parts while suppressing off-box regions. (2) *Scoring.* IoU enforces geometric consistency with the box, Cover penalizes off-box leakage, and CLIP helps disambiguate candidates without altering the given class $c$. Per-metric min–max normalization calibrates scales so $(\alpha, \beta, \gamma)$ are comparable. (3) *Polygonization & clipping.* Douglas–Peucker simplifies boundaries; clipping to $B$ guarantees consistency with the source box; degenerate cases ($< 3$ vertices) fall back to the rectangle. (4) *Compatibility.* The emitted YOLO-Polygon preserves $c$ and is directly usable in second stage transfer. Our bbox2poly generation can be found in Algorithm 1.

## 3.6 UNIFIED OPTIMIZATION

We use SGD with momentum and L2 weight decay, together with a short warm-up followed by a linear learning-rate decay (from an initial rate $\eta_0$ to a final rate $\eta_{\text{final}}$ over 30 epochs), and a YOLO-style multi-task loss:

$$\mathcal{L} = \lambda_{\text{box}} \text{CIoU} + \lambda_{\text{cls}} \text{BCE} + \lambda_{\text{dfl}} \text{DFL} + \lambda_{\text{mask}} \mathcal{L}_{\text{mask}}, \tag{8}$$

$$\mathcal{L}_{\text{mask}} = \frac{1}{\sum_i s_i^*} \sum_i \frac{s_i^*}{A_i} \sum_{p \in \Omega(b_i^*)} \text{BCE}\big(\hat{m}_i(p),\ m_i(p)\big). \tag{9}$$

$s_i^*$ objectness score, $A_i$ box area, $p$ pixel, $b_i^*$ GT box, $m_i, \hat{m}_i$ GT/pred. masks.

Objectness-/area-normalized mask loss balances small/large instances. Freezing P5 reduces memory (enabling larger batch and stability), while the RF PRIOR (Eq. 2) preserves detector semantics during transfer.

## 4 EXPERIMENTS

### 4.1 DATASETS AND EXPERIMENT SETUP

*Datasets.* Three splits are derived from the SIDEGUIDE traffic-scene corpus (Park et al., 2020), all sharing the same 35 object categories. (i) *BBox–DS* is a randomly sampled, down-scaled version of the original bounding-box annotations (max side 512 px); it provides high-level context priors for the detector. (ii) *PolySeg* contains pixel-accurate polygon labels converted to YOLO-style masks. (iii) *SurfaceMask* consists of coarse surface masks for road-layout understanding, integrated with the class indices as *PolySeg* so results are directly comparable for Instance Segmentation. We additionally introduce the *Fixed Objects* dataset and categorize all four datasets along two axes—Background (BG) and Label–Space (LS) shift—relative to *BBox–DS*, our prior source. (1) **Background (BG) shift**—changes in scene/background statistics or capture context; (2) **Label-Space (LS) shift**—changes in the category vocabulary or mask type of label.

*PolySeg* keeps the same 35 classes as *BBox–DS* but switches from boxes to polygons; the mask–space supervision and per-image region differences induce a *moderate LS* and *mild BG* shift. *SurfaceMask* is polygon-based with

Table 1: Summary of Datasets. We define datasets as Background (BG) and Label-Space (LS) shift.

| Datasets (#images) | Custom of SIDEGUIDE & New Dataset | | | | |
|---|---|---|---|---|---|
| | Dataset Type | cls | *train set* | *val. set* | *total* |
| BBox–DS (Park et al., 2020) | Prior Source | 29 | 20,097 | 2,233 | 22,330 |
| PolySeg (Park et al., 2020) | BG shift ▲, LS shift ▲ | 29 | 66,082 | 7,342 | 73,424 |
| SurfaceMask (Park et al., 2020) | BG shift ✓, LS shift ✓ | 6 | 41,759 | 4,640 | 46,399 |
| Fixed Objects (Proposed) | BG shift ▲, LS shift ✓ | 15 | 13,577 | 1,543 | 15,120 |

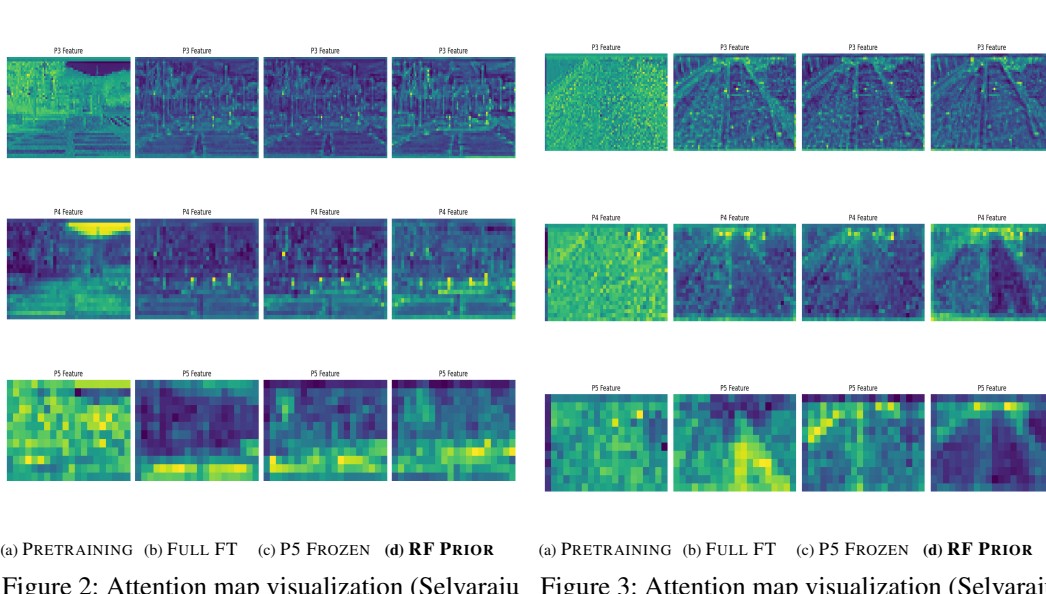

(a) PRETRAINING (b) FULL FT (c) P5 FROZEN **(d) RF PRIOR**     (a) PRETRAINING (b) FULL FT (c) P5 FROZEN **(d) RF PRIOR**

Figure 2: Attention map visualization (Selvaraju et al., 2017) of **PolySeg** sample for each model's trained backbone, $\theta_{bb}$. The mismatch energy $\mathcal{E} = \sum_k \|F_{5,k} - \phi(F_{4/3,k})\|^2$ (Eq. 3.4), is minimized when attention collapses to the frozen P5 context, as visualized.

Figure 3: Attention map visualization (Selvaraju et al., 2017) of **SurfaceMask** for each model's trained backbone, $\theta_{bb}$. The mismatch energy $\mathcal{E} = \sum_k \|F_{5,k} - \phi(F_{4/3,k})\|^2$ (Eq. 3.4), is minimized when attention collapses to the frozen P5 context, as visualized.

surface-centric labels that differ from *BBox–DS*; although the domain is traffic scenes, images emphasize road surfaces, yielding *strong LS* and *noticeable BG* shift. **Fixed Objects (Proposed)** shares the pedestrian/crosswalk background with *BBox–DS/PolySeg* but uses a different label set focused on fixed infrastructure, thus showing a *clear LS* and *small BG* shift (Table 1 summarizes these relations and Appendix§A.2 formalizes it).

*Implementation details.* A YOLOv12-x backbone with an information-separated design is used: Models are pre-trained for 500 epochs with image size $640 \times 640$, batch size 8, using an RTX Quadro A6000 (24 GB & fine-tuning for SIDEGUIDE). and A100 (40GB & fine-tuning for *Fixed Objects*) in mixed precision. All other hyper-parameters follow Ultralytics defaults.

*Evaluation protocol.* We report precision, recall, COCO-style $mAP_{50}$ and $mAP_{50:90}$ for boxes, and mask mAP for polygons. We report the final-epoch score and the per-epoch mean ± std.dev. to track early-stage adaptation within 30, 50 epochs and additional adaptation steps.

### 4.2 ABULATION STUDIES AND PRIOR VISUALIZATION

Experiments are organized as follows:

(1) **No Pretraining:** Segmentation training from scratch. (2) **Full Fine-tuning:** No parameters frozen in $\ell_{seg}$ learning, with transferring pretrained BBox-DS, $\theta_{det.bb}$. (3) **P5 Frozen:** P1–P4 adaptive, P5 fixed in $\ell_{seg}$ learning, with transferring pretrained BBox-DS, $\theta_{det.bb}$. (4) **Proposed Method (RF PRIOR):** Decoupled backbone, with Multi-Scale Attentive Decoder in $\mathcal{L}_{CC}$. Task is to transfer the non-overlapping, fine-grained polygon cues provided by SIDEGUIDE—namely *PolySeg* and *SurfaceMask*—and *Fixed Objects* into the *BBox-DS* backbone, so that pedestrian-related objects can

Table 2: Performance comparison in Instance Segmentation

| | **#Efficiency** $fw(.)$ | | **POLYSEG** | | **SURFACEMASK** | | mean$_{score}$ gain over BASE* | |
|---|---|---|---|---|---|---|---|---|
| **Models** | **FPS/GFLOPs** | **SRI*** | $mAP_{50}^{val}$ | $mAP_{50:90}^{val}$ | $mAP_{50}^{val}$ | $mAP_{50:90}^{val}$ | *overall$^{val}$* | $mAP^{val}$ |
| YOLOv9-E (Wang et al., 2024) | 19.6/1.24 | 0.58 | 46.56 | 27.62 | **75.59** | 59.68 | 2.90 | 2.13 |
| YOLOv11-X (Khanam et al., 2024) | 36.1/1.60 | 1.38 | 46.01 | 27.41 | 74.64 | 59.22 | 1.15 | 1.39 |
| *YOLOv12-X Backbone* (Tian et al., NeurIPS 2025) | | | | | | | | |
| NO PRETRAINING (base)* | 25.8/1.62 | 1.00 | 43.92 | 26.04 | 74.73 | 59.61 | - | - |
| FULL FINE-TUNING | 25.8/1.62 | 1.00 | 44.25 | 26.22 | 74.07 | 59.31 | −1.27 | −0.72 |
| P5 FROZEN | 25.9/1.62 | 1.01 | 42.95 | 25.32 | 74.14 | 59.58 | −0.60 | −0.60 |
| **RF PRIOR (OURS)** | **25.2/3.93** | **2.34** | **48.42** | **28.25** | 75.15 | **60.15** | **3.52** | **3.19** |

Table 3: Performance comparison of box (B) and mask (M) metrics on val. set; last-epoch and second-best results.

| Models | $P$ (B) | $R$ (B) | $mAP_{50}^{val}$ (B) | $mAP_{50:90}^{val}$ (B) | $P$ (M) | $R$ (M) | $mAP_{50}^{val}$ (M) | $mAP_{50:90}^{val}$ (M) |
|---|---|---|---|---|---|---|---|---|
| **SIDEGUIDE FOR DETECTION AND INSTANCE SEGMENTATION** (Park et al., 2020) | | | | | | | | |
| v9-E (Wang et al., 2024) | **71.09**$_{70.20}$ | 49.85$_{50.09}$ | 55.17$_{54.97}$ | 41.91$_{41.72}$ | **69.69**$_{69.49}$ | 46.97$_{46.64}$ | 51.15$_{51.30}$ | 33.25$_{33.15}$ |
| v11-X (Khanam et al., 2024) | 66.08$_{68.82}$ | 47.89$_{47.68}$ | 53.93$_{53.47}$ | 40.97$_{40.58}$ | 67.64$_{66.95}$ | 45.39$_{45.59}$ | 50.58$_{50.15}$ | 33.03$_{32.83}$ |
| NO PRETRAINING | 66.61$_{66.28}$ | 47.15$_{46.65}$ | 52.30$_{52.04}$ | 39.51$_{39.31}$ | 65.41$_{64.72}$ | 44.70$_{44.64}$ | 49.08$_{48.86}$ | 32.08$_{31.95}$ |
| ∟ (50 epochs) | 67.95$_{67.62}$ | 49.61$_{49.95}$ | 54.72$_{54.57}$ | 41.84$_{41.71}$ | 69.54$_{69.27}$ | 45.66$_{45.83}$ | 51.25$_{51.12}$ | 33.53$_{33.47}$ |
| FULL FINE-TUNING | 69.38$_{68.91}$ | 44.97$_{44.88}$ | 51.53$_{51.15}$ | 38.78$_{38.42}$ | 67.72$_{66.85}$ | 43.11$_{43.01}$ | 48.30$_{47.92}$ | 31.48$_{31.22}$ |
| P5 FROZEN | 63.58$_{64.10}$ | 46.77$_{46.09}$ | 51.61$_{51.26}$ | 38.99$_{38.71}$ | 63.38$_{66.56}$ | 44.04$_{43.48}$ | 48.41$_{48.13}$ | 31.57$_{31.36}$ |
| **RF PRIOR** | 70.22$_{69.48}$ | **51.55**$_{51.49}$ | **56.95**$_{56.63}$ | **42.09**$_{41.84}$ | 68.08$_{67.95}$ | **48.91**$_{48.08}$ | **52.86**$_{52.56}$ | **33.81**$_{33.71}$ |
| ∟ (50 epochs) | **75.84**$_{74.52}$ | **56.69**$_{57.08}$ | **63.61**$_{63.21}$ | **48.26**$_{47.96}$ | **75.11**$_{73.96}$ | **52.99**$_{52.87}$ | **58.98**$_{58.66}$ | **37.87**$_{37.64}$ |

Table 4: Performance comparison of box (B) and mask (M) metrics on val. set; mean ± standard deviation (percentage points) over all epochs.

| Models | $P$ (B) | $R$ (B) | $mAP_{50}^{val}$ (B) | $mAP_{50:90}^{val}$ (B) | $P$ (M) | $R$ (M) | $mAP_{50}^{val}$ (M) | $mAP_{50:90}^{val}$ (M) |
|---|---|---|---|---|---|---|---|---|
| **SIDEGUIDE FOR DETECTION AND INSTANCE SEGMENTATION** (Park et al., 2020) | | | | | | | | |
| v9-E (Wang et al., 2024) | **65.53**$_{\pm7.20}$ | 42.09$_{\pm8.54}$ | 46.72$_{\pm9.98}$ | **34.52**$_{\pm8.31}$ | **64.59**$_{\pm6.73}$ | 40.15$_{\pm8.04}$ | 43.94$_{\pm9.19}$ | **28.05**$_{\pm6.31}$ |
| v11-X (Khanam et al., 2024) | 64.67$_{\pm7.05}$ | 40.13$_{\pm7.84}$ | 45.08$_{\pm9.62}$ | 33.15$_{\pm8.03}$ | 63.95$_{\pm6.81}$ | 38.24$_{\pm7.41}$ | 42.38$_{\pm8.93}$ | 27.19$_{\pm6.24}$ |
| NO PRETRAINING | 64.93$_{\pm6.54}$ | 39.01$_{\pm7.39}$ | 43.86$_{\pm9.21}$ | 32.16$_{\pm7.71}$ | 63.88$_{\pm6.22}$ | 37.29$_{\pm6.99}$ | 41.38$_{\pm8.57}$ | 26.51$_{\pm6.01}$ |
| ∟ (50 epochs) | 64.43$_{\pm7.28}$ | 39.34$_{\pm10.85}$ | 48.03$_{\pm14.19}$ | 32.08$_{\pm12.25}$ | 65.08$_{\pm7.17}$ | 37.06$_{\pm9.68}$ | 44.45$_{\pm13.26}$ | 25.30$_{\pm10.60}$ |
| FULL FINE-TUNING | 62.68$_{\pm7.62}$ | 35.71$_{\pm7.67}$ | 40.53$_{\pm9.71}$ | 29.44$_{\pm8.06}$ | 61.59$_{\pm7.31}$ | 34.22$_{\pm7.34}$ | 38.16$_{\pm9.01}$ | 24.34$_{\pm6.33}$ |
| P5 FROZEN | 63.33$_{\pm6.14}$ | 38.82$_{\pm7.47}$ | 43.11$_{\pm9.08}$ | 31.53$_{\pm7.57}$ | 63.03$_{\pm6.11}$ | 36.98$_{\pm7.04}$ | 40.52$_{\pm8.42}$ | 25.97$_{\pm5.91}$ |
| **RF PRIOR** | 64.86$_{\pm6.44}$ | **42.68**$_{\pm8.19}$ | **47.60**$_{\pm9.54}$ | 34.19$_{\pm7.80}$ | 64.23$_{\pm5.92}$ | **40.16**$_{\pm7.50}$ | **44.27**$_{\pm8.71}$ | 27.99$_{\pm6.00}$ |
| ∟ (50 epochs) | **68.20**$_{\pm6.50}$ | **47.13**$_{\pm8.46}$ | **52.58**$_{\pm9.66}$ | **38.51**$_{\pm8.12}$ | **67.16**$_{\pm5.90}$ | **44.34**$_{\pm7.81}$ | **48.89**$_{\pm8.87}$ | **31.08**$_{\pm6.04}$ |

be segmented and detected with higher fidelity. Earlier studies on task transfer focused on COCO-scale pre-training, progressive fine-tuning, or freezing all layers before a chosen stage (Vazquez et al., 2025; Gandhi & Gandhi, 2025). In contrast, we examine a subtler, label-level shift: how performance changes within a similar domain when annotation granularities and object types differ, rather than when the entire visual domain changes. To this end, we design alignment-aware learning schemes that explicitly account for spatial mis-alignment and category mismatches between the two label spaces.

## 4.3 RESULTS

Table 2 reports performance for each segmentation sub-task. To show that a small architectural tweak can still transfer well, we introduce the Speed-Retention Index (SRI). We define $\text{SRI}_i = (F_i \times G_i)/(^*F_{\text{base}} \times G_{\text{base}})$. Our custom head raises GFLOPs ($G$; We report GFLOPs scaled by $10^{-2}$) by $2.4\times$ yet reaches an SRI of 2.34, holding FPS ($F$) loss below 3 % thanks to the memory-bound regime and higher TensorCore utilization. Thus, it improves accuracy while maintaining real-time inference (25 FPS) on an NVIDIA A100 system. RF PRIOR achieves the top mAP on both PolySeg and SurfaceMask. Compared with a 50-epoch, target-focused fine-tune, it raises mAP(50–90), confirming that our Maximum-A-Posteriori formulation boosts accuracy without cross-dataset bottlenecks. Figures 2, 3, 5, 6 illustrate why: in PolySeg, class-specific polygon cues emerge coherently across P3–P5, while in SurfaceMask the model attends to separable polygon lines within the difficult road-surface class over the same feature levels, aligning low- and high-level evidence for stronger segmentation. Details under varying training conditions are provided in Figures 7, 8, 9.

### 4.3.1 DETECTION AND SEGMENTATION ON SIDEGUIDE

On SIDEGUIDE (Tables 3–4), all rows except those explicitly marked "(50 epochs)'' are trained for 30 epochs. Under this budget, RF PRIOR delivers the strongest last-epoch performance across detection and instance segmentation, reaching $\text{mAP}_{50}^{\text{val}}$=**56.95 / 52.86** and $\text{mAP}_{50:90}^{\text{val}}$=**42.09 / 33.81** for B/M, respectively, which clearly exceeds *No Pretraining* (52.30/49.08 and 39.51/32.08) and *Full*

Table 5: Performance comparison of box (B) and mask (M) metrics on val. set; Top three rows show last-epoch and second-best results. Second rows show mean ± standard deviation over all epochs.

| Models | $P$ (B) | $R$ (B) | $mAP^{val}_{50}$ (B) | $mAP^{val}_{50:90}$ (B) | $P$ (M) | $R$ (M) | $mAP^{val}_{50}$ (M) | $mAP^{val}_{50:90}$ (M) |
|---|---|---|---|---|---|---|---|---|
| **FIXED OBJECTS FOR DETECTION AND INSTANCE SEGMENTATION** | | | | | | | | |
| Adaptation steps=16.98k, A2C2f scale=1.2 (Tian et al., 2025) | | | | | | | | |
| NO PRETRAINING | $79.77_{76.98}$ | $64.28_{62.06}$ | $72.00_{68.26}$ | $52.54_{49.32}$ | $69.54_{66.48}$ | $55.01_{54.47}$ | $59.14_{57.76}$ | $40.45_{38.38}$ |
| ⌐ (w/o AreaAtten.) | $79.38_{78.14}$ | $66.72_{64.60}$ | $73.44_{71.72}$ | $53.57_{50.83}$ | $73.46_{70.32}$ | $55.78_{55.60}$ | $62.27_{60.51}$ | $42.98_{40.48}$ |
| **RF PRIOR** | $\mathbf{80.26}_{75.58}$ | $66.50_{67.02}$ | $\mathbf{75.62}_{73.07}$ | $\mathbf{55.21}_{53.16}$ | $69.70_{66.90}$ | $\mathbf{57.23}_{55.56}$ | $\mathbf{63.58}_{60.45}$ | $\mathbf{43.71}_{41.36}$ |
| NO PRETRAINING | $62.85_{\pm15.94}$ | $48.50_{\pm16.10}$ | $51.64_{\pm20.84}$ | $35.24_{\pm16.37}$ | $\mathbf{58.47}_{\pm10.26}$ | $41.61_{\pm14.21}$ | $43.72_{\pm16.73}$ | $28.17_{\pm11.96}$ |
| ⌐ (w/o AreaAtten.) | $61.98_{\pm16.71}$ | $49.54_{\pm17.43}$ | $52.40_{\pm21.81}$ | $34.95_{\pm16.76}$ | $57.10_{\pm14.25}$ | $42.96_{\pm14.89}$ | $44.90_{\pm18.08}$ | $28.85_{\pm12.81}$ |
| **RF PRIOR** | $\mathbf{64.19}_{\pm16.06}$ | $\mathbf{51.23}_{\pm17.20}$ | $\mathbf{54.80}_{\pm21.51}$ | $\mathbf{37.06}_{\pm16.85}$ | $57.96_{\pm12.99}$ | $\mathbf{44.47}_{\pm14.35}$ | $\mathbf{46.64}_{\pm17.38}$ | $\mathbf{30.19}_{\pm12.60}$ |
| Adaptation steps=50.94k | | | | | | | | |
| NO PRETRAINING | $\mathbf{87.41}_{86.49}$ | $73.83_{74.14}$ | $83.36_{83.57}$ | $67.01_{66.70}$ | $\mathbf{76.53}_{77.66}$ | $63.64_{64.19}$ | $70.30_{71.33}$ | $51.23_{51.48}$ |
| ⌐ (w/o AreaAtten.) | $83.72_{81.40}$ | $81.29_{81.60}$ | $\mathbf{86.64}_{85.98}$ | $69.55_{68.67}$ | $73.54_{70.91}$ | $70.84_{70.92}$ | $73.51_{73.15}$ | $\mathbf{53.86}_{53.77}$ |
| **RF PRIOR** | $83.99_{84.24}$ | $\mathbf{80.13}_{79.89}$ | $86.48_{86.08}$ | $\mathbf{69.84}_{69.38}$ | $72.96_{72.81}$ | $\mathbf{69.58}_{69.19}$ | $73.05_{72.64}$ | $53.68_{53.47}$ |
| NO PRETRAINING | $\mathbf{77.06}_{\pm13.68}$ | $63.44_{\pm14.18}$ | $70.13_{\pm17.85}$ | $52.98_{\pm16.03}$ | $\mathbf{68.72}_{\pm9.52}$ | $54.70_{\pm12.50}$ | $59.12_{\pm14.70}$ | $41.06_{\pm11.67}$ |
| ⌐ (w/o AreaAtten.) | $75.88_{\pm13.80}$ | $65.84_{\pm15.66}$ | $72.03_{\pm18.86}$ | $54.17_{\pm17.05}$ | $67.55_{\pm11.07}$ | $57.09_{\pm13.59}$ | $61.31_{\pm15.75}$ | $42.75_{\pm12.58}$ |
| **RF PRIOR** | $76.00_{\pm13.05}$ | $\mathbf{66.05}_{\pm15.31}$ | $\mathbf{72.34}_{\pm18.35}$ | $\mathbf{54.54}_{\pm16.75}$ | $67.02_{\pm10.30}$ | $\mathbf{57.15}_{\pm13.09}$ | $61.23_{\pm15.10}$ | $42.82_{\pm12.25}$ |

*Fine-Tuning* (51.53/48.30 and 38.78/31.48). Notably, *P5 Frozen* already matches or slightly surpasses full fine-tuning at the early stage (51.61/48.41 vs. 51.53/48.30 in mAP$_{50}$ for B/M; 38.99/31.57 vs. 38.78/31.48 in mAP$_{50:90}$), supporting the view that preserving large-RF semantics while adapting lower stages accelerates alignment under BG/LS shift. Averaged over epochs (Table 4), RF PRIOR again shows the highest means with moderate variance: for B it attains $\mathbf{47.60} \pm 9.54$ (mAP$_{50}$) and $\mathbf{34.19}\pm7.80$ (mAP$_{50:90}$); for M, $\mathbf{44.27}\pm8.71$ and $\mathbf{27.99}\pm6.00$. Both *No Pretraining* (43.86±9.21 / 32.16±7.71 for B; 41.38±8.57 / 26.51±6.10 for M) and *Full Fine-Tuning* (40.45±9.71 / 29.44±8.06; 38.16±9.01 / 24.34±6.33) lag behind. Extending only RF PRIOR to 50 epochs further lifts the means to $52.58 \pm 9.60/38.51 \pm 8.12$ (B) and $48.89 \pm 8.87/31.08 \pm 6.04$ (M), indicating faster and more stable convergence than training from scratch—even when the latter is given a longer schedule (cf. *No Pretraining* at 50 epochs: 54.72/41.84 for B and 51.25/33.53 for M). Finally, while *YOLOv9-E* exhibits strong precision ($P(B) = 71.09$, $P(M) = 69.69$) consistent with its gradient-concentration design, its attention adaptation under shift remains limited; with the same budget, RF PRIOR attains higher mask mAP$_{50}$ (52.86 vs. 51.15) without relying on additional throughput-oriented tweaks.

## 5 DISCUSSION

**Fixed Objects (Table 5 & Figure 4).** Because the FIXED OBJECTS is substantially smaller than SIDEGUIDE, we down-scaled the A2C2f backbone by a factor of 1.2 to avoid overfitting. Even under this tighter capacity budget, RF PRIOR adapts more effectively than the *No Pretraining* and is more stable than RF PRIOR w/o AreaAtten. throughout the early adaptation window.

**Early budget (16.98k steps, A2C2f scale=1.2).** At the same compute budget shown at the top of Table 5, RF PRIOR achieves **P(B)=80.26₇₅.₅₈**, **R(B)=66.50₆₇.₀₂**, **mAP$^{val}_{50}$(B)=75.62₇₃.₀₇**, **mAP$^{val}_{50:90}$(B)=55.21₅₃.₁₆**, and on masks

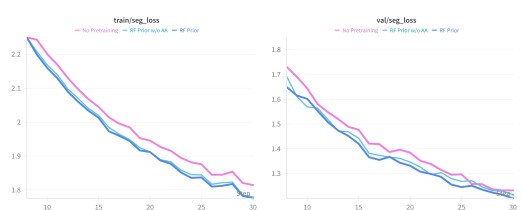

Figure 4: **Left:** Target Adaptation loss (train) and **Right:** Target validation risk in FIXED OBJECTS.

**P(M)=69.70₆₆.₉₀**, **R(M)=57.23₅₅.₅₆**, **mAP$^{val}_{50}$(M)=63.58₆₀.₄₅**, **mAP$^{val}_{50:90}$(M)=43.71₄₁.₃₆**. These numbers improve upon *No Pretraining* ( e.g., **72.00₆₈.₂₆** mAP$^{val}_{50}$(B) and **59.14₅₇.₇₆** mAP$^{val}_{50}$(M) ) by **+3.62** B-mAP$_{50}$ and **+4.44** M-mAP$_{50}$, and over RF PRIOR W/O AREAATTEN. (**73.44₇₁.₇₂ / 62.27₆₀.₅₁**) by **+2.18** (B) and **+1.31** (M) absolute mAP$_{50}$ points. Figure 4 corroborates this: both the training and validation seg_loss curves for RF PRIOR sit below the others from the earliest steps onward, and the corresponding mAP rises sooner in the adaptation trajectory.

Table 6: Training and validation loss deltas between RF PRIOR and RF PRIOR (w/o AreaAtten.) across 299.6k adatptation steps in SIDEGUIDE. We use $\Delta = $ RF PRIOR $-$ RF PRIOR (w/o AreaAtten.). "Last step" is 299.6k. "Global", "Early" (0–149.8k), and "Late" (149.8k–299.6k) report the mean $\pm$ standard deviation of step-wise deltas over those ranges. "Best step" marks where the most negative $\Delta$ occurs. 14,978 steps $\approx$ one full pass over the training set. Additionally, we report the mean log-delta (mean $\Delta_{\log}\pm$std) and the geometric-mean ratio ($\downarrow$%) across 0–299.6k steps; positive $\downarrow$% indicates average percentage reduction in loss for RF PRIOR.

| Loss | Last step | Last $\Delta$ | Global mean±std | Early mean±std | Late mean±std | Best step | Best $\Delta$ | Mean $\Delta_{\log}$±std | GM ratio ($\downarrow$%) |
|---|---|---|---|---|---|---|---|---|---|
| train/box_loss | 299.6k | -0.10968 | $-0.25817 \pm 0.17441$ | $-0.37392 \pm 0.18366$ | $-0.14241 \pm 0.02662$ | 15.0k | -0.76653 | $-0.19280 \pm 0.06036$ | **17.54%** |
| train/seg_loss | 299.6k | -0.27932 | $-0.37709 \pm 0.08485$ | $-0.44165 \pm 0.07239$ | $-0.31253 \pm 0.02643$ | 15.0k | -0.60440 | $-0.17359 \pm 0.01635$ | **15.94%** |
| train/cls_loss | 299.6k | -0.22018 | $-0.48829 \pm 0.31038$ | $-0.69579 \pm 0.32502$ | $-0.28080 \pm 0.04546$ | 15.0k | -1.37547 | $-0.31002 \pm 0.06431$ | **26.66%** |
| train/dfl_loss | 299.6k | -0.19145 | $-0.36693 \pm 0.21945$ | $-0.50646 \pm 0.24016$ | $-0.22741 \pm 0.02700$ | 15.0k | -0.99273 | $-0.24601 \pm 0.07340$ | **21.81%** |
| val/box_loss | 299.6k | -0.03425 | $-0.23336 \pm 0.25246$ | $-0.40081 \pm 0.26650$ | $-0.06591 \pm 0.03500$ | 15.0k | -0.97990 | $-0.19155 \pm 0.14490$ | **17.43%** |
| val/seg_loss | 299.6k | -0.00963 | $-0.19094 \pm 0.20008$ | $-0.34185 \pm 0.17992$ | $-0.04003 \pm 0.03913$ | 15.0k | -0.61513 | $-0.09980 \pm 0.08731$ | **9.50%** |
| val/cls_loss | 299.6k | -0.02323 | $-0.37914 \pm 0.48071$ | $-0.67547 \pm 0.53793$ | $-0.08282 \pm 0.05776$ | 15.0k | -1.92284 | $-0.25264 \pm 0.20725$ | **22.33%** |
| val/dfl_loss | 299.6k | -0.08400 | $-0.24929 \pm 0.22282$ | $-0.39044 \pm 0.24463$ | $-0.10815 \pm 0.02653$ | 15.0k | -0.93564 | $-0.18350 \pm 0.11495$ | **16.76%** |

**Larger budget (50.94k steps).** The gains persist when we triple the adaptation steps (bottom half of Table 5). RF PRIOR reaches about **86.5** B-mAP$_{50}$ and **73.1** M-mAP$_{50}$ with strong precision/recall (*e.g.*, $\approx$ 84/80 on boxes and $\approx$ 73/70 on masks). The second rows in each block report the epoch with the second-best checkpoint, and the bottom rows report the mean±std over all checkpoints; in both summaries RF PRIOR remains at least competitive on boxes and consistently better on masks. We attribute the slightly larger variance on FIXED OBJECTS to genuine domain shift: while BBox-DS and outdoor backgrounds overlap with the training distribution, *indoor* backgrounds appear in the validation set only. This out-of-distribution shift inflates variance relative to SIDEGUIDE, yet RF PRIOR still secures higher mAP early and maintains a safe margin over *No Pretraining* (Table 5, Figure 4). Compared to RF PRIOR W/O AREAATTEN., the loss traces show a more consistent gap to *No Pretraining*, suggesting that AreaAtten. dampens oscillations from the indoor/outdoor mixture.

**SideGuide (Table 6).** We summarizes stepwise loss deltas at equal budgets, using $\Delta = $ RF PRIOR $-$ RF PRIOR (w/o AreaAtten.). Across the full 0–299.6k steps, RF PRIOR reduces losses relative to its ablation: *train*—box_loss ($\Delta_{\log} = -0.1928\pm0.0604$, GM ratio $\downarrow$**17.54%**), seg_loss ($-0.1736\pm0.0163$, $\downarrow$**15.94%**), cls_loss ($-0.3100\pm0.0643$, $\downarrow$**26.66%**), dfl_loss ($-0.2460\pm0.0734$, $\downarrow$**21.81%**); *validation*—box_loss ($-0.1916\pm0.1449$, $\downarrow$**17.43%**), seg_loss ($-0.0998\pm0.0873$, $\downarrow$**9.50%**), cls_loss ($-0.2526\pm0.2073$, $\downarrow$**22.33%**), dfl_loss ($-0.1835\pm0.1150$, $\downarrow$**16.76%**). The most negative deltas ("best step") consistently occur around **15.0k** steps, indicating that AreaAtten. not only lowers the average risk but also accelerates early adaptation—precisely the regime of interest for rapid deployment.

**Takeaways.** (i) With a smaller backbone on FIXED OBJECTS, RF PRIOR already yields better mAP at low budgets while keeping losses lower and steadier (Figure 4); (ii) variance is higher than SIDEGUIDE due to indoor backgrounds unique to validation, yet the method preserves its early-step advantage; (iii) on SIDEGUIDE, the quantitative loss analysis shows clear, *systematic* reductions over the ablation across all four losses, confirming that RF PRIOR is both sample-efficient and robust to target shift. Accordingly, we will extend this line of work across a broader spectrum of real-world datasets, toward a more stable and sustainable source–target framework.

## 6 CONCLUSION

We proposed a simple, effective strategy for efficient instance segmentation transfer by explicitly preserving global receptive-field priors derived from detection tasks. Our method significantly enhances segmentation performance and training efficiency by freezing the deepest, globally-contextualized block (P5) with alining decoder. This approach bridges detection and segmentation tasks effectively, presenting a practical transfer learning strategy adaptable to various multi-task vision frameworks.

ETHICS STATEMENT

*Scope and intended use.* Our work studies representation reuse and weakly supervised instance segmentation. The method is designed for research and benchmarking with generating real-domain polygon; but it is *not* safety-certified for safety-critical systems (e.g., autonomous driving) in paper. Any deployment must include additional hazard analyses, on-road testing, and compliance audits.

*Data provenance and consent.* All experiments use publicly available traffic-scene images and labels (or our own annotations) that do not intentionally include personally identifiable information (PII). We do not attempt to infer sensitive attributes (e.g., identity, race, health). If any image incidentally contains PII (faces, license plates), we apply standard obfuscation or omit such samples from release. We respect the original dataset licenses and terms of use; any redistributed annotations follow those licenses.

*Weak-label mining risks.* Our BBox→Polygon conversion may propagate dataset or foundation-model biases (e.g., class vocabulary biases from CLIP, proposal biases from SAM). To reduce this risk, our pipeline (i) gates masks by detector-aligned geometry (IoU/coverage), (ii) limits the class set to non-sensitive, utilitarian object categories, and (iii) clips polygons to the annotated box to discourage off-target leakage. We recommend auditing pseudo-labels before downstream use and avoiding sensitive categories.

*Annotator welfare and credit.* If new annotations (e.g., *Fixed Objects*) are released, annotators are to be trained with clear guidelines, credited in documentation, and compensated according to institutional policies. We avoid collecting harmful content and provide a takedown contact for data subjects.

*Reproducibility and transparency.* We plan to release full training code, configuration files, bbox2polygon.py, and scripts to regenerate all figures/tables, along with documentation of dataset splits and any post-processing. This aims to facilitate independent verification, error reporting, and responsible reuse by the community.

REPRODUCIBILITY STATEMENT

The code implementing our method should be released upon publication. We provide all the necessary details to reproduce our experiments in the Section 4 and in the Appendix§A.

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

## A APPENDIX

### A.1 BACKGROUND AND RELATED WORK

**Representation reuse and anchored adaptation.** A central problem in transfer learning is *how to adapt a pretrained representation without erasing its invariances.* Early evidence established that low/mid-level conv features transfer broadly while high-level features become task-specific (Yosinski et al., 2014). From an optimization view, existing approaches can be organized by *what is allowed to move* relative to the source solution $\theta_0$:

$$\textbf{(A)} \quad \min_{\boldsymbol{\theta}} \; \mathcal{L}_{\text{tgt}}(\boldsymbol{\theta}) + \tfrac{\lambda}{2} \|\boldsymbol{\theta} - \boldsymbol{\theta}_0\|_2^2 \tag{10}$$

$$\textbf{(B)} \quad \min_{\phi} \; \mathcal{L}_{\text{tgt}}(\boldsymbol{\theta}_0, \phi) \tag{11}$$

$$\textbf{(C)} \quad \min_{\boldsymbol{\theta}_{\text{train}}} \; \mathcal{L}_{\text{tgt}}\big(\boldsymbol{\theta}_{\text{train}}, \boldsymbol{\theta}_{\text{frozen}} := \boldsymbol{\theta}_{0,\text{frozen}}\big) \tag{12}$$

*Notes.* (A) Anchored regularization / L2-SP. (B) PEFT with a frozen backbone and small trainables (adapters/LoRA/BitFit/VPT(Houlsby et al., 2019; Hu et al., 2022; Ben Zaken et al., 2022; Jia et al., 2022)). (C) Hard-freeze subsets used in practice for stability and efficiency.

(A) keeps the whole model plastic but contracts it toward $\boldsymbol{\theta}_0$, mitigating destructive drift and improving stability/conditioning (Xuhong et al., 2018). (B) makes the contraction *implicit* by clamping the backbone and only learning a small set of parameters; this is parameter-efficient and often robust when target data are scarce (Houlsby et al., 2019; Hu et al., 2022). (C) is widely used in large vision models (e.g., freezing deepest blocks or early ViT stages) to preserve global attention patterns while specializing shallower components. Our method sits at the intersection: we impose a *receptive-field–aware prior* that *freezes* the largest-RF block and applies Gaussian shrinkage to the remaining blocks, yielding a precise *MAP* objective

$$\min_{\boldsymbol{\theta}_3, \boldsymbol{\theta}_4, \boldsymbol{\theta}_{\text{seg}}} \; \mathcal{L}_{\text{tgt}}(\boldsymbol{\theta}_3, \boldsymbol{\theta}_4, \boldsymbol{\theta}_5 = \boldsymbol{\theta}_{0,5}) + \tfrac{\xi_3}{2} \|\boldsymbol{\theta}_3\|_2^2 + \tfrac{\xi_4}{2} \|\boldsymbol{\theta}_4\|_2^2, \tag{13}$$

which is equivalent to a block-diagonal Gaussian prior on trainable blocks and a delta prior on the frozen block (Sec. 3.2). This *scale-coupled* view differs from generic PEFT/anchoring in two ways. First, the prior is aligned with the architecture's multi-scale semantics: the deepest block (stride-32, largest RF) encodes global scene structure and is preserved exactly, while mid/low-RF blocks are regularized but plastic. Second, we prove that this choice induces a *global-to-local* gradient pathway: error signals traverse the frozen high-RF features and concentrate updates onto small-RF parameters where boundary and texture cues matter most (Appendix§A.3).

Dense prediction pipelines already exploit representation reuse across scales. FPN and Mask R-CNN propagate high-level semantics downward (Lin et al., 2017; He et al., 2017), while modern backbones (Swin/ConvNeXt) improve the quality of these multiscale features (Liu et al., 2021; 2022). Decoder designs such as DETR/Mask2Former inject global reasoning but do so at limited fusion sites to control cost (Carion et al., 2020; Cheng et al., 2022). We adopt the same philosophy: a *single* global-to-local attention site couples the frozen global template to adaptable mid/low-resolution maps (Sec. 3.3). Analytically, restricting global attention to one site reduces complexity from $O(N^2)$ to $O(N^2/a)$ with area partitioning while retaining long-range cues where they have the highest leverage (mid scales). Empirically, this design prevents wholesale retuning of global semantics and focuses capacity on spatial details, matching the theoretical picture given by our gradient-flow analysis.

**Weak supervision from boxes and automatic polygon mining.** When dense masks are limited, bounding boxes provide a strong but coarse prior. Classic methods refine boxes into masks via proposal mining and consistency regularization (Dai et al., 2015; Khoreva et al., 2017); BoxInst shows that instance segmentation is learnable from boxes alone with alignment losses (Tian et al., 2021). Large pretrained models further strengthen this conversion pipeline: CLIP supplies class-consistency scores from text–image alignment (Radford et al., 2021), and SAM yields high-quality, box-prompted proposals at scale (Kirillov et al., 2023). We leverage these capabilities but enforce *detector consistency* end-to-end: (i) generate multiple SAM masks per box; (ii) gate by IoU/coverage to reject off-box leakage; (iii) optionally re-rank by CLIP to prefer semantically on-class candidates; (iv) simplify contours with Douglas–Peucker and *clip* polygons to the original box (Sec. 3.5). This choice is not merely heuristic—under our *MAP* objective, the weak labels and the adaptation bias are *mutually reinforcing*. The frozen deepest block provides a stable global template; polygon priors sharpen local residuals; and the single attention site transmits these residuals as targeted updates to $P3/P4$. In contrast to PEFT that introduces extra trainables or to box-supervised pipelines that treat label mining as a separate pre-processing step, our conversion is tied to the detector's geometry and to the RF-aware prior, closing the loop between *what the model preserves*, *where it learns*, and *how weak labels are shaped*.

**Contrast to prior art.** Prior work typically (i) regularizes toward $\boldsymbol{\theta}_0$, (ii) freezes most weights and learns small adapters, or (iii) mines masks from boxes. We combine all three perspectives coherently: a scale-aware *MAP* prior (freeze largest-RF block, shrink others), a *single* global-to-local attention

site for efficient context injection, and a detector-aligned SAM+CLIP polygon miner whose outputs are geometrically constrained. Our analysis (Appendix§A.3) explains why this combination yields localized, boundary-focused updates while preserving global semantics, aligning theory with practice.

## A.2 Dataset Taxonomy and Shift Formalization

**Setup.** Let the prior source be $\mathcal{S}$ with joint $P_{\mathcal{S}}(x, y)$, label set $C_{\mathcal{S}}$, and supervision type $\tau_{\mathcal{S}} \in$ $\{\texttt{box}, \texttt{polygon}, \texttt{surface}\}$. Any target dataset $\mathcal{D}$ has $P_{\mathcal{D}}(x, y)$, $C_{\mathcal{D}}$, $\tau_{\mathcal{D}}$.

**BG shift (covariate/context).** We measure background change by any nonnegative divergence between *marginals*:

$$d_{\text{BG}}(\mathcal{D} \mid \mathcal{S}) := \mathsf{D}\big(P_{\mathcal{D}}(x) \,\|\, P_{\mathcal{S}}(x)\big), \qquad \mathsf{D} \in \{\text{KL}, \chi^2, \text{IPM}, \text{W}_2, \ldots\}. \tag{14}$$

**LS shift (label space / supervision granularity).** Let the vocabulary distance be the Jaccard complement and the supervision mismatch be a simple indicator:

$$d_{\text{cls}}(\mathcal{D} \mid \mathcal{S}) := 1 - \frac{|C_{\mathcal{D}} \cap C_{\mathcal{S}}|}{|C_{\mathcal{D}} \cup C_{\mathcal{S}}|}, \qquad d_{\text{sup}}(\mathcal{D} \mid \mathcal{S}) := \mathbf{1}[\tau_{\mathcal{D}} \neq \tau_{\mathcal{S}}]. \tag{15}$$

Combine them as a single score

$$d_{\text{LS}}(\mathcal{D} \mid \mathcal{S}) := d_{\text{cls}}(\mathcal{D} \mid \mathcal{S}) + \kappa \, d_{\text{sup}}(\mathcal{D} \mid \mathcal{S}), \qquad \kappa \in [0, 1]. \tag{16}$$

**Dataset type (decision rule).** For thresholds $\delta_{\text{BG}}, \delta_{\text{LS}} > 0$,

$$\text{Type}(\mathcal{D} \mid \mathcal{S}) = \begin{cases} \text{PRIOR SOURCE}, & d_{\text{BG}} < \delta_{\text{BG}} \,\wedge\, d_{\text{LS}} < \delta_{\text{LS}}, \\ \text{BG SHIFT}, & d_{\text{BG}} \geq \delta_{\text{BG}} \,\wedge\, d_{\text{LS}} < \delta_{\text{LS}}, \\ \text{LS SHIFT}, & d_{\text{BG}} < \delta_{\text{BG}} \,\wedge\, d_{\text{LS}} \geq \delta_{\text{LS}}, \\ \text{BG + LS SHIFT}, & d_{\text{BG}} \geq \delta_{\text{BG}} \,\wedge\, d_{\text{LS}} \geq \delta_{\text{LS}}. \end{cases}$$

It depends only on the underlying distributions and simple set relations. In our experiments we treat $\mathcal{S}$=*BBox–DS* and choose $(\delta_{\text{BG}}, \delta_{\text{LS}}, \kappa)$ on a validation split; the qualitative assignments in Table 1 follow directly from the definitions in equation 14–equation 16.

## A.3 Why Freezing High-RF Features Yields Global-to-Local Gradient Flow

Let the backbone produce $F_3, F_4, F_5$ at strides $(8, 16, 32)$ with radii $R_3 < R_4 < R_5$, and decoder $Y = H_{\boldsymbol{\theta}_{\text{seg}}}(F_3, F_4, F_5)$. We train

$$\min_{\boldsymbol{\theta}_3, \boldsymbol{\theta}_4, \boldsymbol{\theta}_{\text{seg}}} \frac{1}{N} \sum_{i=1}^{N} \ell\big(H_{\boldsymbol{\theta}_{\text{seg}}}(F_3^i, F_4^i, F_5^i), y_i\big) \quad \text{s.t.} \quad \boldsymbol{\theta}_5 = \boldsymbol{\theta}_5^0, \tag{17}$$

with $F_5 = B_5(F_4; \boldsymbol{\theta}_5^0)$, $F_4 = B_4(F_3; \boldsymbol{\theta}_4)$, $F_3 = B_3(x; \boldsymbol{\theta}_3)$.

**Assumption 1** (Locality and regularity). *$B_\ell$ are CNN blocks with finite RF radii $R_\ell$, piecewise $C^1$ in their arguments; $H_{\boldsymbol{\theta}_{\text{seg}}}$ is $C^1$ in its inputs.*

**Gradients still pass through the frozen block.** By chain rule, for any sample (drop the index $i$),

$$\nabla_{\boldsymbol{\theta}_4} \ell = \Big( \frac{\partial \ell}{\partial F_4} + \underbrace{\frac{\partial \ell}{\partial F_5} \frac{\partial F_5}{\partial F_4}}_{\text{via frozen } B_5} \Big) \frac{\partial F_4}{\partial \boldsymbol{\theta}_4}, \qquad \nabla_{\boldsymbol{\theta}_3} \ell = \frac{\partial \ell}{\partial F_3} \frac{\partial F_3}{\partial \boldsymbol{\theta}_3}, \tag{18}$$

where $\frac{\partial F_5}{\partial F_4} = J_{5 \leftarrow 4}(F_4; \boldsymbol{\theta}_5^0)$ is a *fixed* Jacobian. Thus, although $\nabla_{\boldsymbol{\theta}_5} \ell = 0$ (frozen), gradients *propagate through* $B_5$ *to* $F_4$ *and then to* $\boldsymbol{\theta}_4$.

**Assumption 2** (Non-degenerate coupling). *There exists $\sigma_{\min} > 0$ such that the smallest singular value of $J_{5 \leftarrow 4}$ at training points is $\geq \sigma_{\min}$ (i.e., $B_5$ does not collapse $F_4$ to a constant along training trajectories).*

**Proposition 1** (Lower bound on update signal). *Under Assumptions 1–2, $\big\| \nabla_{\boldsymbol{\theta}_4} \ell \big\| \geq \sigma_{\min} \big\| \frac{\partial \ell}{\partial F_5} \frac{\partial F_4}{\partial \boldsymbol{\theta}_4} \big\|$.*

*Proof.* From equation 18, $\frac{\partial \ell}{\partial F_5} J_{5 \leftarrow 4}$ is a nonzero linear form unless $\frac{\partial \ell}{\partial F_5} = 0$; its operator norm is bounded below by $\sigma_{\min} \| \frac{\partial \ell}{\partial F_5} \|$. Multiply by $\frac{\partial F_4}{\partial \boldsymbol{\theta}_4}$ and take norms. $\qquad \square$

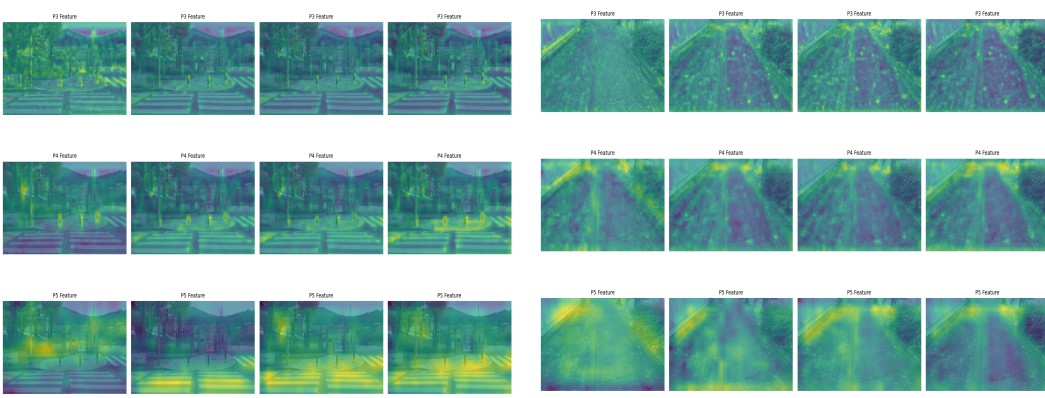

(a) Pretraining  (b) Full FT  (c) P5 Frozen  **(d) RF Prior**   (a) Pretraining  (b) Full FT  (c) P5 Frozen  **(d) RF Prior**

Figure 5: Attention map (overlaid on the original image) visualization (Selvaraju et al., 2017) of **PolySeg** sample for each model's trained backbone, $\boldsymbol{\theta}_{\text{bb}}$. The mismatch energy $\mathcal{E} = \sum_k \|F_{5,k} - \phi(F_{4/3,k})\|^2$ (Eq. 3.4), is minimized when attention collapses to the frozen P5 context.

Figure 6: Attention map (overlaid on the original image) visualization (Selvaraju et al., 2017) of **SurfaceMask** for each model's trained backbone, $\boldsymbol{\theta}_{\text{bb}}$. The mismatch energy $\mathcal{E} = \sum_k \|F_{5,k} - \phi(F_{4/3,k})\|^2$ (Eq. 3.4), is minimized when attention collapses to the frozen P5 context.

**Why updates are spatially localized.** Let $w^{(\ell)}$ be a convolutional kernel in $B_\ell$, shared over spatial sites. Denote by $\Omega^{(\ell)}(w)$ the set of output locations whose computation involves $w^{(\ell)}$. Then

$$\frac{\partial \ell}{\partial w^{(\ell)}} = \sum_{(u,v) \in \Omega^{(\ell)}(w)} \frac{\partial \ell}{\partial z^{(\ell)}(u,v)} \, x^{(\ell-1)}\big(u + \Delta_u, \, v + \Delta_v\big), \tag{19}$$

where $z^{(\ell)}$ is the pre-activation and $(\Delta_u, \Delta_v)$ is within the kernel's spatial support. By locality, $\Omega^{(3)}(w)$ covers many *small* RF footprints in the image; $\Omega^{(4)}(w)$ covers fewer but larger footprints; and $\Omega^{(5)}(w)$ spans coarse, near-global footprints. When $\boldsymbol{\theta}_5$ is frozen, $\frac{\partial \ell}{\partial w^{(5)}} \equiv 0$, eliminating global, coarse-grained adjustments. Error reduction must therefore occur via $w^{(3)}$ and $w^{(4)}$, whose supports correspond to *local* neighborhoods. Consequently, parameter updates affect the prediction predominantly within unions of $\mathcal{N}_{R_3}$ and $\mathcal{N}_{R_4}$ around high-loss sites, yielding *boundary-focused* corrections.

**Assumption 3** (Decoder coupling). $H_{\boldsymbol{\theta}_{\text{seg}}}$ *fuses* $(F_3, F_4, F_5)$ *with a locally Lipschitz attention/projection* $\phi$ *into the* $F_5$ *space (query/key-value), and the training loss is* $\beta$-*smooth in the fused features.*

Define the alignment energy

$$E(\Phi) = \sum_k \big\|F_{5,k} - \phi\big(F_{4,k};\Phi\big)\big\|_2^2 + \big\|F_{5,k} - \phi\big(F_{3,k};\Phi\big)\big\|_2^2, \tag{20}$$

where $k$ indexes spatial locations and $\Phi$ collects the decoder's projection/attention parameters.

**Lemma 1** (Descent of alignment energy). *If $E$ is $L$-smooth in $\Phi$, then with step size $\eta \in (0, 1/L]$, $E(\Phi^+) \leq E(\Phi) - \frac{\eta}{2}\|\nabla_\Phi E(\Phi)\|_2^2$.*

*Proof.* Standard smoothness (Descent Lemma). □

**Theorem 1** (Global-to-local gradient flow). *Under Assumptions 1–3, freezing $\boldsymbol{\theta}_5$ yields: (i) non-vanishing gradient signals to $\boldsymbol{\theta}_4$ (Prop. 1); (ii) updates that are confined to unions of RF neighborhoods determined by $R_3, R_4$; (iii) monotone reduction of the alignment mismatch equation 20 for suitable steps on decoder parameters, which in turn induces localized corrections in $F_4, F_3$ that better agree with the fixed global template $F_5$.*

*Proof.* (i) follows from Prop. 1. (ii) follows from convolutional locality and the fact that $\boldsymbol{\theta}_5$ cannot change. (iii) follows from Lemma 1; the gradient $\nabla_\Phi E$ backpropagates to $\boldsymbol{\theta}_4, \boldsymbol{\theta}_3$ through $\phi$ and $B_4, B_3$, but the target $F_5$ stays fixed, so corrections occur at sites $k$ with large residuals $F_{5,k} - \phi(F_{\ell,k})$, i.e., near boundaries/high-error regions. $\square$

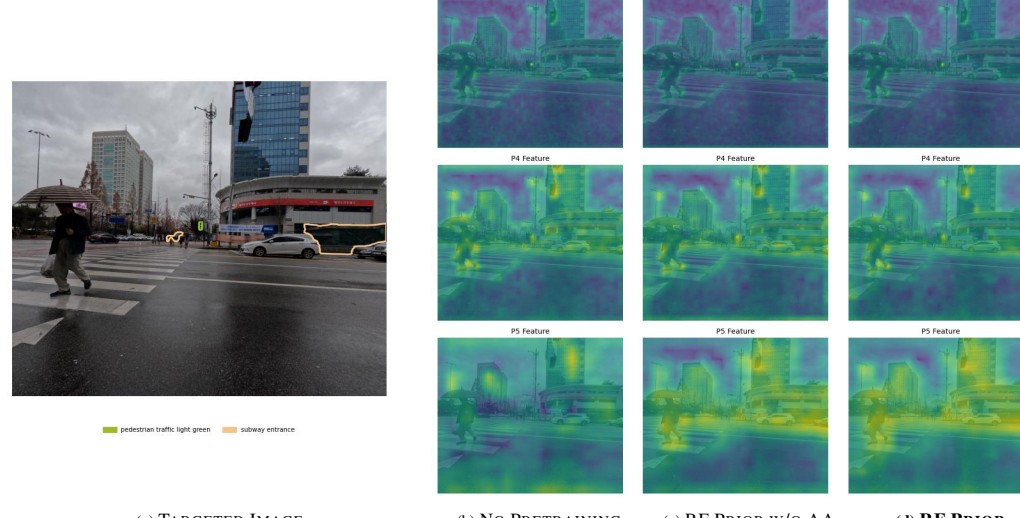

(a) TARGETED IMAGE      (b) NO PRETRAINING      (c) RF PRIOR W/O AA      (d) **RF PRIOR**

Figure 7: Attention map (overlaid on the original image) visualization (Selvaraju et al., 2017) of **Fixed Objects** sample for each model's trained backbone, $\boldsymbol{\theta}_{\mathrm{bb}}$. The mismatch energy $\mathcal{E} = \sum_k \|F_{5,k} - \phi(F_{4/3,k})\|^2$ (Eq. 3.4), is minimized when attention collapses to the frozen P5 context. RF Prior steers global prior knowledge toward local objects through attention; with AREAATTENTION in our MULTI-SCALE ATTENTIVE DECODER, the global–local mismatch decreases and the backbone outputs become more compact compared to w/o AREAATTENTION.

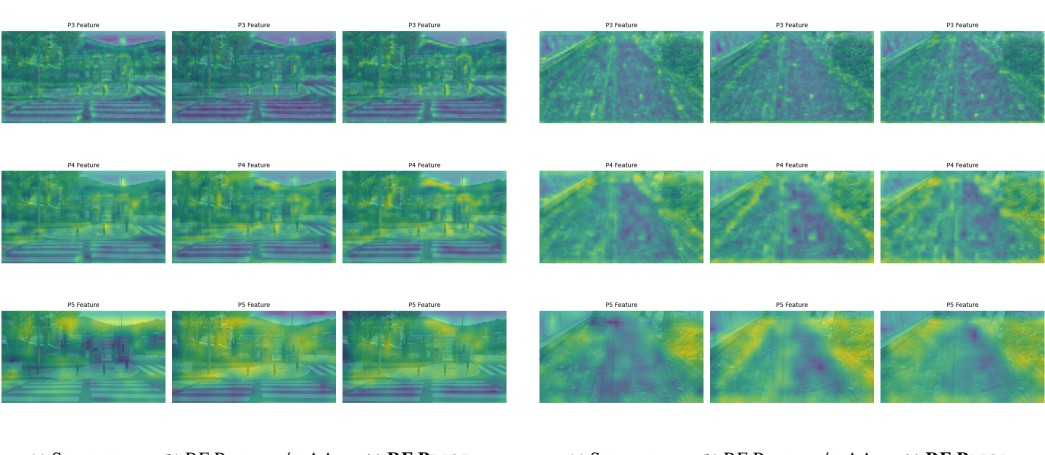

(a) SCRATCH    (b) RF PRIOR W/O AA    (c) **RF PRIOR**      (a) SCRATCH    (b) RF PRIOR W/O AA    (c) **RF PRIOR**

Figure 8: Attention map (overlaid on the original image) visualization (Selvaraju et al., 2017) of **PolySeg** sample for each model's tuned (**Fixed Objects**) backbone, $\boldsymbol{\theta}_{\mathrm{bb}}$. We define these phenomenon as smooth modulation of LS shift.

Figure 9: Attention map (overlaid on the original image) visualization (Selvaraju et al., 2017) of **SurfaceMask** for each model's tuned (**Fixed Objects**) backbone, $\boldsymbol{\theta}_{\mathrm{bb}}$. We define these phenomenon as smooth modulation of BS + LS shift.

**Algorithm 1** Automatic BBox-to-Polygon Conversion

---

**Require:** Detector-trained image sets with box labels; SAM predictor $P$; optional CLIP scorer $C$; merge strategy $m \in \{\texttt{best}, \texttt{union}\}$ with IoU threshold $\tau$; weights $\alpha, \beta, \gamma$; polygon tolerance $\varepsilon$; visualization limit $N_{\text{viz}}$.

**Ensure:** For each image, a polygon label file (YOLO-polygon format).

1: **for all** dataset directory $d$ in $\texttt{train\_dirs}$ **do**
2: $\quad$ $images \leftarrow$ all files in $d/\texttt{images}$
3: $\quad$ create folder $d/\texttt{polygon\_labels}$ if it does not exist
4: $\quad$ **for all** image $I$ in $images$ with size $H \times W$ **do**
5: $\quad\quad$ load $I$ and its YOLO label file $L$; **continue** if $L$ missing
6: $\quad\quad$ $P.\text{set\_image}(I)$
7: $\quad\quad$ $polys \leftarrow [\,]$
8: $\quad\quad$ **for all** label line $l \in L$ **do** $\qquad\qquad\qquad$ ▷ $l = (\texttt{class}, c_x, c_y, w, h)$ in YOLO format
9: $\quad\quad\quad$ $(c, B) \leftarrow \text{YOLOToXYXY}(l, W, H)$
10: $\quad\quad\quad$ $\{M_k\}_{k=1}^{K} \leftarrow P.\text{predict}(\texttt{box} = B, \texttt{multiMask} = True)$
11: $\quad\quad\quad$ **if** $m = \texttt{union}$ **then**
12: $\quad\quad\quad\quad$ $M^\star \leftarrow \bigvee_{k:\,(M_k, B) \geq \tau} M_k$ $\qquad\qquad\qquad$ ▷ pixelwise OR
13: $\quad\quad\quad\quad$ **if** no $k$ satisfies $(M_k, B) \geq \tau$ **then**
14: $\quad\quad\quad\quad\quad$ $M^\star \leftarrow \arg\max_k (M_k, B)$
15: $\quad\quad\quad\quad$ **end if**
16: $\quad\quad\quad$ **else**
17: $\quad\quad\quad\quad$ **for** $k = 1$ **to** $K$ **do**
18: $\quad\quad\quad\quad\quad$ $_k \leftarrow \dfrac{|M_k \cap B|}{|M_k \cup B|}, \quad \text{Cover}_k \leftarrow \dfrac{|M_k \cap B|}{|M_k|}$
19: $\quad\quad\quad\quad\quad$ $s_k^{\text{clip}} \leftarrow \begin{cases} C.\text{score}(I, M_k, \texttt{class\_name}(c)), & \text{if } C \text{ exists} \\ 0, & \text{otherwise} \end{cases}$
20: $\quad\quad\quad\quad$ **end for**
21: $\quad\quad\quad\quad$ normalize $_k, s_k^{\text{clip}}, \text{Cover}_k$ to $[0,1]$: $\widetilde{\phantom{i}}_k, \widetilde{s_k^{\text{clip}}}, \widetilde{\text{Cover}_k}$
22: $\quad\quad\quad\quad$ $k^\star \leftarrow \arg\max_k \left( \alpha \widetilde{\phantom{i}}_k + \beta \, \widetilde{s_k^{\text{clip}}} + \gamma \, \widetilde{\text{Cover}_k} \right)$
23: $\quad\quad\quad\quad$ $M^\star \leftarrow M_{k^\star}$
24: $\quad\quad\quad$ **end if**
25: $\quad\quad\quad$ $poly \leftarrow \text{MASKToPolygon}(M^\star, \varepsilon)$
26: $\quad\quad\quad$ **if** $poly$ is empty **or** $|poly| < 3$ **then**
27: $\quad\quad\quad\quad$ $poly \leftarrow$ rectangle corners of $B$
28: $\quad\quad\quad$ **end if**
29: $\quad\quad\quad$ $poly \leftarrow \text{CLIPPolygon}(poly, B)$
30: $\quad\quad\quad$ $coords \leftarrow \text{NORMALIZEPolygon}(poly, W, H)$
31: $\quad\quad\quad$ append string "$c$ $coords$" to $polys$
32: $\quad\quad\quad$ **if** $\texttt{viz\_dir}$ specified **and** #debug_images $< N_{\text{viz}}$ **then**
33: $\quad\quad\quad\quad$ draw $poly$ on a copy of $I$ and push to debug list
34:
35: $\quad\quad\quad\quad$ write $polys$ to $d/\texttt{polygon\_labels}$ (file name matches $I$)
36:
37:
38: $\quad\quad\quad\quad$ **if** $\texttt{viz\_dir}$ specified **and** debug list not empty **then**
39: $\quad\quad\quad\quad\quad$ save montage to $\texttt{viz\_dir}/montage.jpg$
40: $\quad\quad\quad\quad$ **end if**

---

