# OpenReview forum: "RF Prior: Preserving Global-Context Priors for Efficient Instance Segmentation Transfer"
_ICLR.cc/2026/Conference — ICLR 2026 Conference Withdrawn Submission_

### Official Review · Reviewer_wYSu · 2025-10-29

**Soundness:** 2
**Presentation:** 1
**Contribution:** 2
**Rating:** 2
**Confidence:** 4

**Summary:**

This paper proposes a transfer learning framework for instance segmentation that reuses pretrained detector backbone (specifically YOLOv12) by freezing the deepest block (P5) to preserve global context. The approach is formulated as a MAP objective with a block-diagonal Gaussian prior, and includes a multi-scale attentive decoder and a box-to-polygon conversion pipeline using SAM and CLIP. While the idea of preserving receptive field priors is conceptually sound and the empirical results show improvements in segmentation accuracy and training stability, the method is optimised for traffic-scene datasets and lacks validation across diverse domains. Additionally, the polygon generation relies on heuristic combinations that are not fully ablated, and comparisons are limited to YOLO-based baselines, weakening the broader impact.

**Strengths:**

1. The paper presents a clear MAP-based formulation for transfer learning, combining hard freezing of the deepest backbone block with regularized fine-tuning of mid-level layers. While conceptually sound, this builds on existing practices like weight decay and freezing, offering a structured but not fundamentally new approach.
2. Freezing the largest receptive field block (P5) to preserve global semantics is a reasonable design choice, supported by empirical improvements in early-stage adaptation.
3. The decoder integrates area-restricted attention and stride-preserving pooling to fuse multi-scale features efficiently. This helps maintain real-time performance, though the added architectural complexity may pose challenges for reproducibility and deployment in lightweight settings.
4. The integration of SAM and CLIP for converting bounding boxes into polygon masks is practical and aligns well with the detector’s geometry. While useful in weakly supervised scenarios, the pipeline relies on heuristics whose robustness is not fully explored.

**Weaknesses:**

1. The paper’s core idea such as preserving global-context priors via receptive field-aware freezing, builds on well-established techniques such as layer freezing and L2 regularization. The novelty lies mainly in their combination and interpretation, rather than introducing fundamentally new mechanisms. Thus, the motivation feels incremental, and the paper would benefit from a clearer justification of how its approach meaningfully advances beyond prior transfer learning strategies like adapter-based PEFT or partial freezing.
2. All experiments are conducted on traffic-scene datasets. The method’s applicability to other domains (e.g., indoor scenes, medical imaging, aerial imagery) is not tested, raising concerns about generalizability.
3. The approach assumes that the frozen P5 block encodes transferable global semantics. This assumption may not hold under significant domain or task shifts, especially when the target task requires different high-level abstractions.
4. The decoder includes multiple components (SPPF, A2C2f, gated attention) that increase architectural complexity. While efficient in FLOPs, the added design complexities may limit reproducibility and deployment in lightweight or real-time systems.
5. The box-to-polygon conversion pipeline relies on heuristic combinations of IoU, CLIP scores, and contour simplification. The impact of these choices is not fully ablated, and the robustness of the pipeline under noisy or ambiguous inputs is unclear.
6. Comparisons are restricted to YOLO-family models. The absence of evaluations against stronger segmentation baselines (e.g., SegFormer, Mask2Former, SAM2) limits the ability to justify the performance gains.

**Questions:**

1. Could the authors improve the clarity of the write-up and more explicitly highlight the novel aspects of their approach compared to existing PEFT and transfer learning strategies?
2. Can you provide evidence or justification for its applicability to other domains such as indoor environments, medical imaging, or aerial imagery, where global context and object semantics may differ significantly?
3. The approach relies on freezing the P5 block to preserve global semantics. How do you ensure that the frozen features remain relevant across tasks with different high-level abstractions or semantic structures, and have you tested scenarios where this assumption breaks down?
4. The polygon generation pipeline relies on heuristic combinations of IoU thresholds, CLIP scores, and contour simplification. Can you provide a detailed ablation or sensitivity analysis to justify these design choices and demonstrate the robustness of the pipeline under noisy or ambiguous bounding box inputs?

---

### Official Review · Reviewer_AQ5A · 2025-10-31

**Soundness:** 2
**Presentation:** 2
**Contribution:** 2
**Rating:** 4
**Confidence:** 3

**Summary:**

The paper presents an efficient transfer-learning framework that reparameterizes a state-of-the-art detector backbone for polygon-based instance segmentation. It provides a rigorous prior design paradigm for vision model transfer. To utilize box-only datasets under background/label shift, a simple mining module is proposed: it transforms detector boxes into polygon pseudo-masks via candidate segmentation, multi-metric ranking and contour simplification. Integrating these pseudo-masks into the RF-prior pipeline improves boundary metrics significantly with minimal to no inference overhead.

**Strengths:**

1、 This study deconstructs the core contradiction between "preserving priors" and "task adaptation" in model fine-tuning. By leveraging a differentiated framework of "freezing P5 to anchor global semantics + locally adapting P3-P4" and Maximum A Posteriori (MAP) theoretical modeling, it provides a rigorous prior design paradigm for vision model transfer.

2、The visualization analysis of feature maps (Attention map visualization in Figure 2 and 3) and experimental results demonstrate the effectiveness of the proposed method.

**Weaknesses:**

1、Both the method design and method validation in this paper are based on YOLO-12. Is this too restrictive, resulting in a lack of proof for generalization ability? Could additional experiments be added to demonstrate generalization?

2、Why is validation not conducted on some general segmentation datasets such as COCO? Additionally, the proposed dataset seems to make no significant contributions in terms of indicators like the number of categories and scale.

3、The references in the paper propose several box-to-polygon methods (e.g., SAM) as well as some commonly used perception methods based on VLMs. Please explain the advantages of the paper continuing to adopt YOLO-based methods. Furthermore, it is well-known that YOLO-based methods are fast, but the paper seems to not provide timeliness metrics.

**Questions:**

See weekness part.

---

### Official Review · Reviewer_2b2W · 2025-10-31

**Soundness:** 2
**Presentation:** 1
**Contribution:** 2
**Rating:** 2
**Confidence:** 3

**Summary:**

This paper introduces a transfer-learning framework for polygon-based instance segmentation using a YOLO-family backbone. The key idea is to freeze the largest receptive-field backbone block (P5) to retain global semantic context learned during detection, while fine-tuning intermediate blocks (P3--P4) to improve boundary precision.
The method is formalised as a  MAP optimisation with a block-diagonal Gaussian prior, unifying hard-freeze (delta prior) on P5 with weight decay on adaptable blocks.
A multi-scale attentive decoder aligns global context with local features, and an automatic box-to-polygon generation pipeline (SAM + IoU coverage/CLIP ranking + Douglas–Peucker) further supports weakly supervised adaptation.
Experiments on SIDEGUIDE datasets show improved mAP and faster, more stable convergence than full fine-tuning, naive freezing, or training from scratch.

**Strengths:**

1. Clear motivation: freezing large RF features to preserve global semantics is intuitive and practical.
2. Formal MAP formulation, beyond heuristic freezing.
3. Strong engineering contribution: memory-efficient adaptation, YOLO compatibility, lightweight attention design.
4. Useful box-to-polygon conversion pipeline leveraging SAM + CLIP + geometry constraints.
5. Solid experiments: ablations, convergence analysis, Grad-CAM visualisations.
6. Real-world applicability: robust transfer under background and label-space shift.

**Weaknesses:**

1. The evaluation domain is narrow (traffic datasets only), limiting generality.
2. Gains are meaningful but not dramatic; partly incremental over existing transfer heuristics.
3. Theoretical contribution is light; mostly engineering/architectural refinement.
4. Missing comparison against robust SAM-based fine-tuning baselines.
5. Box-to-polygon quality was not discussed, and only a qualitative result for illustration purposes is shown
6. The computational cost could be discussed more thoroughly.
7. The main problem is, however, the writing and presentation: the math lacks an accurate introduction of terms and a suitable explanation. There are several typos: e.g. "Abulation" in the title of paragraph 4.2.

**Questions:**

1. How does the method perform on canonical benchmarks (COCO, LVIS)?
2. Is the approach architecture-agnostic (e.g., ConvNext, ViT backbones)?
3. How does this compare to elastic weight consolidation or progressive/freezing schedules?
4. What is the overhead of SAM + CLIP polygon mining and sensitivity to ranking weights?
5. Can P5 freezing be annealed over training or made data-dependent?

---

### Official Review · Reviewer_ZPT5 · 2025-11-01

**Soundness:** 2
**Presentation:** 2
**Contribution:** 2
**Rating:** 4
**Confidence:** 2

**Summary:**

The paper proposes an efficient transfer learning framework, RF PRIOR (Receptive-Field Prior), for adapting a pre-trained SOTA detector backbone (instantiated with a YOLO-family model) to polygon-based instance segmentation. The key innovation is a novel inductive bias: freezing the highest-level feature block (P5) to preserve global object context, while allowing the lower layers (P3-P4) and the segmentation head to be fine-tuned for local boundary adaptation. This strategy is theoretically formalized using a block-diagonal Gaussian prior under a MAP objective. The framework also integrates an Attentive Decoder with Area-restricted attention for efficient feature alignment, and a BBox-to-Polygon Mining module. The method demonstrates strong, stable performance and fast convergence compared to baselines.

**Strengths:**

1. **Systematic and complete methodology:** The approach is consistent and interpretable, spanning from theoretical derivations (gradient flow, update dynamics) to module design (attentive decoder, pseudo-label generation).
2. **Performance:** The method achieves competitive results, particularly improving on boundary-related metrics ($mAP_{50}(M)$), and exhibits fast convergence.
3. **Practical:** The method is compatible with the YOLOv12 models, easy to deploy, and includes pseudo-code (Algorithm 1) and a reproducibility commitment.

**Weaknesses:**

1. **Weak Core Novelty:** The core mechanism—hard-freezing the highest semantic block (P5)—can be viewed as a simple empirical choice. While formalized, the novelty of the operation itself is questioned. More compelling experimental evidence is needed to prove its superiority over simpler freezing schemes or existing Parameter-Efficient Fine-Tuning (PEFT) methods.
2. **Limited Generality:** The entire framework (RF PRIOR and Attentive Decoder) appears highly coupled with the specific feature pyramid architecture of the YOLOv12-x models (P3, P4, P5). Its applicability to other widely-used backbones like ResNet, Swin Transformer, or other general detection frameworks remains unproven, severely limiting its perceived utility as a general transfer strategy.
3. **Inadequate Comparisons:** The experimental section is insular. The absence of comparisons with established, non-YOLO instance segmentation methods prevents a fair assessment of its competitiveness against the state-of-the-art.
4. **Presentation and Context:** The writing quality is problematic, which significantly impairs comprehension. The central Figure 1 (Method Overview) is visually chaotic, the flow is disorganized, making the core mechanisms and overall framework structure difficult to follow and potentially misleading to the reader. Furthermore, the paper demonstrates a lack of academic rigor through obvious proofreading errors, such as the misspelling of "Ablation" as "Abulation" in the title of Section 4.2.

**Questions:**

1. **Necessity and Optimality of P5 Freezing:** The paper explains that P5 is frozen to preserve global context, but fails to justify that P5 is the optimal block for this operation. The authors do not explore whether alternative strategies for handling P5 (e.g., fine-tuning P5 while freezing P4, or using L2 regularization on P5) might yield superior results.
2. **Missing Related Work Comparison:** The paper fails to discuss how existing work in cross-task transfer learning (e.g., detection to segmentation) addresses the specific problem tackled by this paper. Clarify how established transfer learning methods handle the detection-to-segmentation challenge and explicitly highlight RF PRIOR's unique advantages.
3. **PEFT Contextualization and Ablation Clarity:** The parameter efficiency of RF PRIOR is not quantitatively compared against mainstream PEFT methods (LoRA/Adapter). Furthermore, crucial implementation details for ablation baselines, such as how "Full Fine-tuning" was precisely executed, are missing.
4. **Architecture Universality:** The method's tight coupling with the YOLO P3-P5 structure limits its generality, with unproven applicability to non-YOLO backbones (ResNet, ViT). Offer a detailed theoretical analysis or a small-scale proof-of-concept experiment on one non-YOLO backbone (e.g., ResNet) to demonstrate the generality of the strategy.
5. **SOTA Instance Segmentation Baseline:** The final model's competitiveness is unclear due to the absence of comparison with leading non-YOLO instance segmentation methods. Supplement the results with a comparison table against SOTA non-YOLO baselines, including performance metrics and efficiency metrics (e.g., throughput/trainable parameters).
6. **Figure 1 and Flow Clarification:** Figure 1 is confusing, and the mechanism of the Attentive Decoder's fusion of frozen P5 (global) and fine-tuned P4 (local) features is ambiguous. Provide a simplified, clear sketch in the rebuttal that unambiguously illustrates the Attentive Decoder's P5/P4 fusion details and explains how Area-restricted Attention ensures efficient, targeted information transfer.
7. **Writing Rigor and Oversight:** The paper exhibits numerous oversights in terms of writing and presentation quality, such as a clear spelling error of "Abulation" instead of "Ablation" in the title of Section 4.2.

---

### Note · Authors · 2025-11-12

I have read and agree with the venue's withdrawal policy on behalf of myself and my co-authors.